# HIGHLY EFFICIENT 8-BIT LOW PRECISION INFERENCE OF CONVOLUTIONAL NEURAL NETWORKS

## ABSTRACT

High throughput and low latency inference of deep neural networks are critical for the deployment of deep learning applications. This paper presents a general technique toward 8-bit low precision inference of convolutional neural networks, including **1)** channel-wise scale factors of weights, especially for depthwise convolution, **2)** Winograd convolution, and **3)** topology-wise 8-bit support. We experiment the techniques on top of a widely-used deep learning framework. The 8-bit optimized model is automatically generated with a calibration process from FP32 model without the need of fine-tuning or retraining. We perform a systematical and comprehensive study on 18 widely-used convolutional neural networks and demonstrate the effectiveness of 8-bit low precision inference across a wide range of applications and use cases, including image classification, object detection, image segmentation, and super resolution. We show that the inference throughput and latency are improved by 1.6X and 1.5X respectively with minimal within 0.6%[1] to no loss in accuracy from FP32 baseline. We believe the methodology can provide the guidance and reference design of 8-bit low precision inference for other frameworks. All the code and models will be publicly available soon.

## 1 INTRODUCTION

While convolutional neural networks (CNN) shows state-of-the-art (SOTA) accuracy for wide range of computation vision tasks, it still faces challenges during industrial deployment due to its high computational complexity of inference. Low precision is one of the key techniques being actively studied recently to conquer the problem Vanhoucke et al. (2011); Hwang & Sung (2014); Rastegari et al. (2016); Miyashita et al. (2016); Mellempudi et al. (2017). With hardware acceleration support, low precision inference can compute more operations per second, reduce the memory access pressure and better utilize the cache, and deliver higher throughput and lower latency.

Convolution is the primary operation in CNN models and it is a common practice to enable 8-bit low precision (INT8) inference for convolution in deep learning frameworks (e.g., TensorFlow, MXNet, and TensorRT). To make it work, convolution utilizes INT8 computation, which requires two scale factors for activation and weight, respectively. It is workable for standard convolution with single group and two groups Krizhevsky et al. (2012). However, it does not work well for convolution with large groups, especially for depthwise convolution Howard et al. (2017); Chollet (2016). In addition to direct convolution, it is worthwhile to explore INT8 Winograd convolution Lavin & Gray (2016) for better performance, which is absent in previous research [2]. Although recent work have demonstrated INT8 inference with minimal accuracy loss across various models Vanhoucke et al. (2011); Gysel (2016); Wu et al. (2016); Jacob et al. (2017), INT8 inference is limited due to more complex topology primarily introduced by sum operation in residual block He et al. (2016) and concatenation operation in inception block Szegedy et al. (2015; 2016). Existing solutions need to convert the convolution output from INT8 to FP32, and apply the sum or concatenation operation on FP32. The sacrifice of memory bandwidth and frequent data conversion lead to considerable performance overhead and therefore limit the real deployment. Moreover, there is no systematical study of INT8 inference on various use cases, including image classification Krizhevsky et al. (2012); Simonyan & Zisserman (2014); Szegedy et al. (2015; 2016); He et al. (2016), object detection Ren

---

[1] % denotes percentage point for simplicity in the context of accuracy loss

[2] http://nvdla.org/primer.html

et al. (2015); Dai et al. (2016); Liu et al. (2016), image segmentation Long et al. (2015); Wei et al. (2016), etc.

In this paper, we present a general technique towards efficient INT8 inference of CNN models. We experiment the technique on top of a widely-used deep learning framework. To the best of our knowledge, our work is the first attempt to address the above problems. We summarize our contributions below:

1. We provide a systematical approach to channel-wise quantization of convolution, which is essential to keep the accuracy for depthwise convolution. Top1 accuracy of INT8 inference on MobileNet-V1 and MobileNet-V2 is improved by 1.98% and 70.6%, respectively.

2. We explore the approach of INT8 Winograd convolution and present the calibration details that cannot be trivially derived from direct convolution. Our experiment on VGG-16 shows Top1 and Top5 accuracy loss with INT8 Winograd convolution is minimal within 0.30% and 0.25% from FP32 baseline, reducing from 5.31% and 3.38%, respectively.

3. We add the support of sum in residual block, concatenation in inception block, and convolution for classification. We also fuse the memory-bound operation convolution with a rectified linear unit (ReLU) Nair & Hinton (2010) and fold the parameters of batch normalization Ioffe & Szegedy (2015) into convolution kernels. With topology-wise INT8 support, inference speed is greatly improved by data conversion reduction and memory saving.

4. To our knowledge, this is the first time such a systematic study is applied to and empirical result is reported on many CNN use cases and models. We develop a calibration tool that automatically generates optimized INT8 model from FP32 model without the need of fine-tuning or retraining for easy and repeatable deployment. We perform a comprehensive study on 18 widely-used CNN models and demonstrate the effectiveness of INT8 inference across a wide range of applications, including image classification, object detection, image segmentation, and super resolution. The inference throughput and latency are improved by 1.6X and 1.5X respectively, while the accuracy loss is minimal within 0.6% to no loss from FP32 baseline.

We believe our methodology is general for CNN models and can provide the guide and reference on other frameworks. All the code and models will be publicly available soon.

The rest of the paper is organized as follows, Section 2 discusses related work on low-precision inference in deep learning. Section 3 describes INT8 inference quantization approach and recipe for CNN models. Section 4 includes experimental results, comprehensive study, and related discussion. Finally, Section 5 concludes the summary with results and future directions.

## 2 RELATED WORK

Computer vision tasks win considerable attentions in deep learning field in recent years. Although CNN models provide SOTA accuracy for various computer vision tasks, it still faces challenges during industrial deployment due to its high computational complexity of inference.

NVidia have demonstrated minimal accuracy loss of INT8 inference on several CNN models for image classification (e.g., ResNet-50, ResNet-152, VGG-16, VGG-19, GoogleNet, AlexNet). With the emerging of CNN models, more topology structures have been proposed to accelerate the performance, e.g., depthwise convolution and inception block. Therefore, it is worthwhile to study INT8 inference on those new structures and understand whether traditional approaches are suitable. Besides TensorRT INT8 solution, some other open-source deep learning frameworks have started to support INT8 inference. TensorFlow approaches the conversion of floating-point arrays of numbers into INT8 representations as a compression problem and provides a graph transformation tool for model conversion with INT8 quantization operators inserted into the transformed graph. Similar to TensorFlow, MXNet also provides a calibration tool to transform FP32 to INT8 model. However, the throughput and latency is not ideal due to considerable data conversion overhead and lack of computation fusion. In our work, we add INT8 support of sum in residual block, concatenation in inception block, and convolution for classification, together with necessary computation folding on convolution and batch normalization and fusion on convolution and ReLU. On the other hand, Winograd convolution Lavin & Gray (2016) is a fast algorithm to speed up the convolution performance. Although it has been

widely studied and used in FP32 training and inference, INT8 Winograd convolution is still not available in mainstream deep learning frameworks. Our work is the first attempt to explore INT8 Winograd convolution carefully and deliver the workable solution to keep the accuracy.

In additional to existing inference tools and frameworks from industry, many researchers have experimented low-precision inference with customized low-bit for activation and weights in deep learning tasks. INT8 activations and weights have been proposed in Vanhoucke et al. (2011), while biases and first layer input are kept with FP32 for the task of speech recognition on CPUs. CNN approximation has been presented Gysel (2016) to perform automatic network quantization and scoring, using different bit-widths for number representation, to find a good balance between compression rate and network accuracy. Baidu researchers [3] have successfully used 8-bits of fixed precision with 1 sign bit, 4-bits for the integer part and 3-bits for the fractional part. Various quantization techniques have been discussed in Sze et al. (2017), showing minimal to no loss at reduced precision while keeping FP32 for the first and last layers. Deep compression with pruning, quantization, and Huffman coding has been worked out to reduce the storage requirement of neural networks significantly without affecting the accuracy, thus making easy for deployment on edge device Han et al. (2015). Moreover, we focus on the efficient inference on commodity servers while others might require special hardware support like FPGA. Of course, some of our insights like calibrating INT8 Winograd can complement others' work as well.

## 3 RECIPE OF INT8 INFERENCE

In this section, we first formulate quantization and de-quantization mathematically and then present the general recipe of INT8 inference.

### 3.1 QUANTIZATION AND DE-QUANTIZATION

We define a quantization function $Q : \mathbb{R}^n \times \mathbb{R} \times \mathbb{N} \mapsto \mathbb{Z}^n \times \mathbb{R}$ in Equation 1 to turn an $n$-dimensional rational tensor $\mathbf{r}$ into an $n$-dimensional integer tensor $\mathbf{z}$ with the scale factor $q$ and bit-precision $p$. Here $n$ could be of arbitrary dimensionality. The function $Round$ is a rounding function approximating a rational tensor with an integer tensor.

$$Q(\mathbf{r}, q, p) = Q_p(\mathbf{r}, q) = Q_{p,q}(\mathbf{r}) = (\mathbf{z}, q), \quad \mathbf{z} = \max(\min(Round(q\mathbf{r}), 2^p - 1), -2^p),$$
$$where \ \mathbf{r} \in \mathbb{R}^n, q \in \mathbb{R}, p \in \mathbb{N}^+, \mathbf{z} \in \mathbb{Z}^n, Round : \mathbb{R}^n \mapsto \mathbb{Z}^n \tag{1}$$

We also define a de-quantization function $D : \mathbb{Z}^n \times \mathbb{R} \mapsto \mathbb{R}^n$ that approximates the rational tensor $\mathbf{r}$ with its quantized form $\mathbf{z}$ in Equation 2.

$$D(\mathbf{z}, q) = D_q(\mathbf{z}) = \frac{\mathbf{z}}{q} = \mathbf{r}' \approx \mathbf{r} \tag{2}$$

We then define $+$ and $\times$ arithmetics on $(\mathbf{z}, q)$ in Equation 3. Here we assume $+$ and $\times$ have already been defined for tensor $\mathbf{r}$ and $\mathbf{z}$, e.g., when they are matrices.

$$(\mathbf{z}_1, q_1) + (\mathbf{z}_2, q_2) = Q_p(D_{q_1}(\mathbf{z}_1) + D_{q_2}(\mathbf{z}_2), \min(q_1, q_2))$$
$$(\mathbf{z}_1, q_1) \times (\mathbf{z}_2, q_2) = (\mathbf{z}_1 \times \mathbf{z}_2, q_1 q_2) \tag{3}$$

In practice, we perform sampling for each activation, weight and bias tensor on the given dataset to get a maximum absolute value $max$ from each tensor and set the scale factor of the tensor as $\frac{2^p - 1}{max}$ where $p$ is the precision of quantization. $p = 8$ is used for all non-negative activation tensors which are mostly true for popular CNN models after batch normalization operations are folded with convolution and ReLU with zero negative slope is fused into convolution Xu et al. (2015). For potentially negative input tensors such as the one for first convolution, the operation falls back to FP32 since the hardware-accelerated INT8 convolution only supports non-negative activations as input (more details refer to Rodriguez et al. (2018)). $p = 7$ is used for weight tensors. Then most activations and weights can be stored with INT8. We employ round-half-to-even as the $Round$ function for best statistical accuracy.

---

[3]https://cdn.oreillystatic.com/en/assets/1/event/258/Benchmarking deep learning inference Presentation.pptx

## 3.2 GENERAL RECIPE

We present the general INT8 recipe for CNN models, including depthwise convolution, Winograd convolution, and topology-wise more INT8 support.

### 3.2.1 DEPTHWISE CONVOLUTION

As a common practice, INT8 convolution uses a single scale factor for each tensor, i.e. one for activation and one for weight respectively. It is workable for standard convolution with single group (e.g., VGG-16, GoogleNet-V1, and ResNet-50) and two groups (e.g., AlexNet). However, it does not perform well for convolution with large groups, especially for depthwise convolution (e.g., MobileNet-V1 Howard et al. (2017), MobileNet-V2 Sandler et al. (2018)). Different than standard convolution, depthwise convolution applies a single filter per each input channel. As a result, a single tensor-wise scale factor for weight is not capable to represent the dynamic data range of each channel effectively. Figure 1 indicates the distribution of the first 10 filters per output channel for standard convolution (a) and depthwise convolution (b). As the partial filter distribution is representative, we omit the demonstration of entire weight tensor distribution.

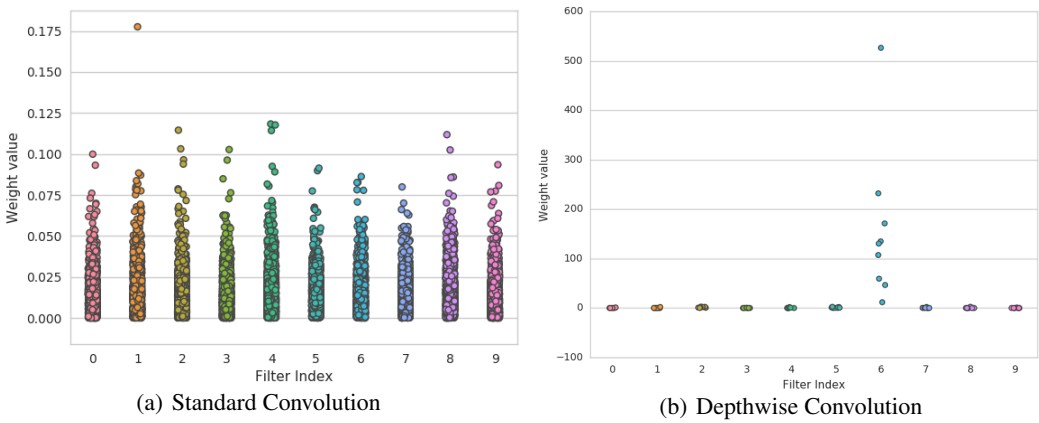

(a) Standard Convolution            (b) Depthwise Convolution

Figure 1: Distribution of the first 10 filters for standard convolution (a) vs. depthwise convolution conv2_1/dwise in MobileNet_V2 (b). Figure (a) shows the stable dynamic data range for the first 10 filters (also for the entire weight tensor). Figure (b) shows the fluctuant dynamic data range cross filters. The 6th filter has 50x bigger dynamic range than the other filters.

Based on the above findings, we propose channel-wise scale factors for weight tensor, similar to Krishnamoorthi (2018). Each scale factor represents the dynamic data range per each filter. The resulting scale factors are $\mathbf{q}_{activation} \times \mathbf{q}_{weight_i}$, where $\mathbf{q}_{activation}$ is the scale factor of activation and $\mathbf{q}_{weight_i}$ is the scale factor of the $i_{th}$ filter. With channel-wise scaling factors, Top1 accuracy of INT8 inference on MobileNet-V1 and MobileNet-V2 is improved by 1.98% and 70.6%, respectively.

### 3.2.2 WINOGRAD CONVOLUTION

Winograd is a fast algorithm for convolution and it has been widely-used in FP32 training and inference Lavin & Gray (2016). However, the study of INT8 Winograd convolution is not publicly available. Considering the attractive performance gains, it is worthwhile to explore INT8 Winograd convolution. We select standard algorithm F(2, 3) for discussion, which can leverage INT8 computation benefit from integer-based input transformation matrix. To make INT8 Winograd convolution work, the key component is to take the scale factor for activation and weight after transformation.

$$\mathbf{x}_a = B^T \, \mathbf{x}_b \, B$$
$$\mathbf{q}_{\mathbf{x}_a} = \mathbf{q}_{\mathbf{x}_b} * max_{\mathbf{x}_b}/max_{\mathbf{x}_a} \tag{4}$$

Equation 4 shows the formula to compute the scale factor after transformation, where $B$ and $B^T$ are transformation matrices defined in Lavin & Gray (2016). Before and after transformation, we have the

activation tensor for $\mathbf{x}_b$ and $\mathbf{x}_a$, the scale factor for $\mathbf{q}_{\mathbf{x}_b}$ (for direction convolution by default) and $\mathbf{q}_{\mathbf{x}_a}$, the maximum absolute value for $max_{\mathbf{x}_b}$ and $max_{\mathbf{x}_a}$, respectively. Similarly, we can compute the scale factor of weight before and after transformation. The scale factor of activation and weight after transformation is set for INT8 Winograd convolution finally. We experiment the idea on VGG-16, a classical model for Winograd convolution. With the scale factor $\mathbf{q}_{\mathbf{x}_a}$, Top1 and Top5 accuracy loss is minimal within 0.30% and 0.25% from FP32 baseline, while with the scale factor $\mathbf{q}_{\mathbf{x}_b}$, the accuracy loss is significant with 5.31% and 3.38%, respectively. Note that our approach is general and can be applied to other algorithms besides standard algorithm F(2, 3).

### 3.2.3 TOPOLOGY-WISE INT8 SUPPORT

We extend INT8 computation to other computation types besides convolution and also apply constant folding and computation fusion to consecutive computations so that almost all input and output activation tensors use INT8 while accumulators use INT32 or FP32 for best accuracy. In this section, we discuss these topology-wise INT8 opportunities. We also discuss topology patterns in which output tensors should be kept in FP32 for good accuracy.

**Pooling.** Both max pooling and average pooling are computed directly with INT8. The scale factors of the input and output tensors are same. We use INT32 accumulator for average pooling to avoid arithmetic overflow.

**Concatenation.** Tensor concatenation is a common operation such as those in Inception block Szegedy et al. (2015; 2016). Figure 2 demonstrates the inception block that concatenates convolution output per filter. Our study shows that the dynamic ranges of the input tensors are quite close. So we set the scale factor of INT8 output tensor to the smallest scale factor of INT8 input tensors.

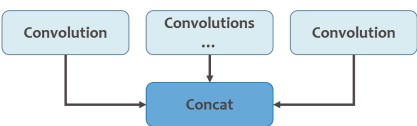

Figure 2: Concatenation in inception block. Dynamic data range is pretty stable in activations for concatenation within the range from 1.6x to 2.0x among the topologies in our work.

**Batch Normalization Folding.** Computing INT8 batch normalization without losing accuracy is challenging. Fortunately, in most recent CNN models, batch normalization is usually added after convolution. Since the computation is essentially an affine transformation during inference, it can be folded into the convolution kernel as in Equation 5. Both the new convolution weight $w'$ and bias $b'$ are affine transformation of the original weight $w$ and bias $b$. As defined in Ioffe & Szegedy (2015), $\mu$ and $\sigma^2$ are the learned mini-batch mean and variance respectively, and $\gamma$ and $\beta$ are the scale and shift terms.

$$w' = (\frac{\gamma}{\sqrt{\sigma^2 + \epsilon}})w, \ \ b' = (\frac{\gamma}{\sqrt{\sigma^2 + \epsilon}})(b - \mu) + \beta. \tag{5}$$

**Fusing Convolution and Element-wise Post-Operations.** For the best arithmetic accuracy and efficient execution, convolution output elements are first accumulated in FP32 and then fused with the element-wise post-operations immediately after it before being quantized back to INT8. The post-operations and quantization can be efficiently computed in registers. Examples of these post-operations are $ReLU$, $Sum$, $Sum\_ReLU$ and $Sum\_BatchNorm\_ReLU$. The latter three are common patterns of residual networks He et al. (2016). Figure 3 illustrates a residual block from ResNet-50 and the sum operation (a) is fused into res2a_branch2c (b). Then, res2a_branch2c accepts two inputs res2a_branch1 and res2a_branch2b, and perform the sum operation. Equation 6 explains the general fused computation $INT8\_Conv\_PostOps$ which computes the INT8 convolution and the post-operations with the INT8 input tensor $\mathbf{z_x}$, INT8 weight tensor $\mathbf{z_w}$, FP32 bias tensor $\mathbf{b}$ and optional arguments $post\_op\_args$ of post-operations. For $Sum\_*$ post-operations, the $post\_op\_args$ is the de-quantized input tensor to add to the convolution output. The quantization precision $p$ is 8 if all elements from $INT8\_Conv\_PostOps$ are positive and resulting data type is unsigned INT8,

otherwise $p$ is 7 for signed INT8 output.

$$(\mathbf{z_y}, q_\mathbf{y}) = Q_{p,q_\mathbf{y}}(INT8\_Conv\_PostOps((\mathbf{z_x}, q_\mathbf{x}), (\mathbf{z_w}, q_\mathbf{w}), \mathbf{b}, post\_op\_args)) \quad (6)$$

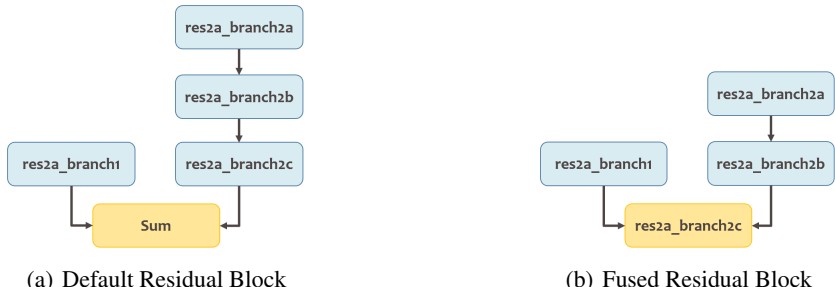

(a) Default Residual Block          (b) Fused Residual Block

Figure 3: Default residual block (a) and fused residual block (b).

**FP32 Tensor for Classification.** Convolution and global average pooling becomes the new operator combination for image classification, which evolves from typical fully-connected operator. Mo-bileNet Howard et al. (2017); Sandler et al. (2018) and SqueezeNet Iandola et al. (2016) are two CNN models with such structure (shown in Figure 4). Our practice shows FP32 tensor output for convolution is essential to keep the accuracy, while using INT8 computation inside convolution. It improves INT8 Top1 accuracy by 6.4% and 1.4% for SqueezeNet-V1_0 and SqueezeNet-V1_1, respectively.

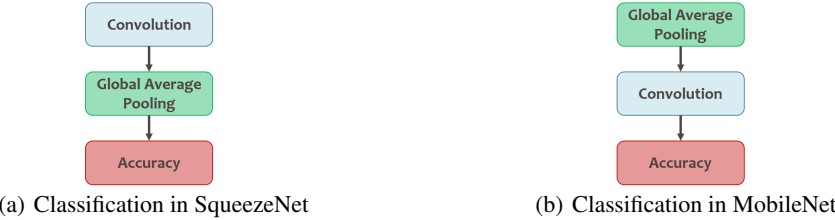

(a) Classification in SqueezeNet          (b) Classification in MobileNet

Figure 4: Classification using convolution and global average pooling.

## 4 EXPERIMENTAL RESULTS

With the general recipe of INT8 inference, we experiment the techniques and develop the calibration tool on top of a widely-used deep learning framework . We next discuss the experimental configurations and perform a systematical study on 18 classical CNN models. We demonstrate the effectiveness of INT8 inference across a wide range of applications and use cases.

### 4.1 CALIBRATION TOOL FOR INT8 INFERENCE

We develop the calibration tool that automatically generates the optimized INT8 model from FP32 model without the need of fine-tuning or retraining. The calibration process has two inputs, CNN model with pre-trained FP32 weights and calibration dataset. Besides, the tool provides the additional items to facilitate the calibration process:

**Iteration number.** It allows user to define the iteration number for sampling on activation.

**Scale factor mode.** It allows user to define scale factor mode single or multiple (channel-wise).

**Calibration strategy.** It allows users to define the calibration algorithm (*Direct* or *KL*) to compute the scale factor by $\frac{2^p-1}{max}$, where $p$ is the quantization precision. *Direct* selects the maximum absolute value of the tensor as $max$ directly, while *KL* computes $max$ in terms of the entropy loss of quantization following the work in TensorRT.

**Accuracy tuning.** It allows users to define the accuracy loss tolerance on INT8 model. Calibration process makes some operations fall back to FP32 to meet the accuracy goal.

## 4.2 EXPERIMENTAL SETUPS

We select totally 18 CNN models in our experiments in Table 1. Basically, we have three rules for model selection: 1) it is classical and representative; 2) it comes from various use cases; and 3) it is publicly available with pre-trained weight or is easy to train with existing hyper-parameters.

Table 1: Selected CNN Models

| Topology | Use Case | Weight | Topology | Use Case | Weight |
|---|---|---|---|---|---|
| VGG-16 | IC | √ | VGG-19 | IC | √ |
| ResNet-50 | IC | √ | ResNet-101 | IC | √ |
| ResNet-152 | IC | √ | ResNet-50 (FB) | IC | × |
| MobileNet-V1 | IC | √ | MobileNet-V2 | IC | √ |
| Inception-V3 | IC | √ | Inception-ResNet-V2 | IC | √ |
| SqueezeNet-V1_0 | IC | √ | SqueezeNet-V1_1 | IC | √ |
| SSD (VGG-16) | OD | √ | SSD (MobileNet-V1) | OD | √ |
| Faster-RCNN (VGG-16) | OD | × | R-FCN (ResNet-101) | OD | × |
| FCN | IS | √ | FSRCNN | SR | × |

Topology column shows the selected CNN model. On ResNet-50, we use two versions, default one from He et al. (2016) and variant one from FaceBook (with FB) Goyal et al. (2017). Use case column shows the model category, IC (image classification), OD (object detection), IS (image segmentation), and SR (super resolution). Weight column shows whether the pre-trained weight is publicly available.

With respect to calibration dataset, we use ImageNet-1k Russakovsky et al. (2015) for image classification, PASCAL VOC Everingham et al. (2015) for object detection and image segmentation, and internal gaming images for super resolution.

## 4.3 ACCURACY AND PERFORMANCE

We perform calibration on training dataset with sampling iteration from 1, 2, 5, 10, 20, to 30, scale factor mode single or multiple, and different algorithm *Direct* or *KL*. The total calibration cost is affordable since it takes seconds to minutes to complete each calibration. We measure the accuracy on validation dataset independently from calibration dataset. Table 2 shows the best accuracy of CNN models under INT8 inference. Note that we use standard metrics to measure the accuracy, Top1 and Top5 for image classification, mAP (mean Average Precision) for object detection, mean accuracy and IoU (Intersection of Union) for image segmentation, and SSIM (Structural SIMilarity) and PSNR (Peak Signal-to-Noise Ratio) for super resolution. Our experiments demonstrate the effectiveness across a wide range of use cases, keeping the accuracy loss from FP32 baseline, within 0.6% for Top1 and 0.3% for Top5 on image classification, 0.5% for mAP on object detection, 0.2% for mean IoU on image segmentation, and 0.1% for PSNR on super resolution. Moreover, INT8 inference recipe also works well for models ResNet-50/101/152 with sparsity removal Rodriguez et al. (2018).

On the other hand, we evaluate the errors of 50k images from ImageNet validation set for FP32 and INT8 inference and find that there is no obvious bias at image class based on empirical analysis on incorrectly-predicted images. With further analysis on typical images, we figure out that it is more difficult for INT8 model to distinguish the objects with small differences. As an example, INT8 model can recognize the dog (ILSVRC2012_val_00046423) correctly, but fails to figure out the accurate breed. Moreover, we find that the information loss from FP32 to INT8 model may lead to potential misclassification (e.g., ILSVRC2012_val_00031193). We also compute the entropy of Softmax output for both FP32 and INT8 model. The results show the probability is average for INT8 model, which indicates the entropy increases and Top1 classification capability decreases.

On performance side, we measure the performance of INT8 inference and speedup over FP32 using dummy data, as shown in Table 2. We can see that the throughput and latency are improved

Table 2: Accuracy and Performance: FP32 VS. INT8

| Topology | Top1 | Top5 | Throughput | Latency |
|---|---|---|---|---|
| VGG-16 | 68.18% (-0.16%) | 88.32% (-0.05%) | 156 (1.5X) | 11.43 (1.3X) |
| VGG-19 | 68.28% (-0.17%) | 88.27% (-0.11%) | 126 (1.5X) | 13.24 (1.3X) |
| ResNet-50 | 72.71% (-0.05%) | 90.97% (-0.04%) | 671 (1.7X) | 2.50 (1.9X) |
| ResNet-101 | 73.92% (-0.10%) | 91.72% (0.00%) | 349 (1.6X) | 6.48 (1.5X) |
| ResNet-152 | 74.48% (-0.27%) | 92.00% (-0.11%) | 234 (1.6X) | 10.44 (1.5X) |
| ResNet-50 (FB) | 76.40% (0.04%) | 93.17% (-0.10%) | 575 (1.7X) | 2.72 (2.1X) |
| MobileNet-V1 | 69.58% (-0.29%) | 89.14% (-0.22%) | 2618 (1.9X) | 0.81 (1.4X) |
| MobileNet-V2 | 71.07% (-0.53%) | 90.00% (-0.24%) | 2042 (2.0X) | 1.43 (1.1X) |
| Inception-V3 | 77.15% (-0.06%) | 93.31% (0.00%) | 393 (1.7X) | 5.33 (1.5X) |
| Inception-ResNet-V2 | 80.11% (-0.49%) | 95.14% (-0.06%) | 163 (1.6X) | 15.27 (1.3X) |
| SqueezeNet-V1_0 | 57.32% (-0.24%) | 80.04% (-0.26%) | 1700 (1.9X) | 0.82 (1.3X) |
| SqueezeNet-V1_1 | 57.85% (-0.19%) | 80.68% (-0.12%) | 2651 (1.8X) | 0.74 (1.0X) |
| SSD (VGG-16)[†] | 77.75% (0.03%) | N/A | 71 (1.5X) | 23.91 (1.6X) |
| SSD (MobileNet-V1)[†] | 72.07% (-0.47%) | N/A | 1038 (1.8X) | 3.69 (2.1X) |
| Faster-RCNN[†] | 71.05% (-0.42%) | N/A | 14 (1.5X) | 72.72 (1.5X) |
| R-FCN[†] | 78.87% (-0.29%) | N/A | 20 (1.6X) | 50.43 (1.6X) |
| FCN[‡] | 82.00% (-0.21%) | 69.73% (-0.11%) | 8 (1.3X) | 128.81 (1.3X) |
| FSRCNN[§] | 0.9000 (-0.0066) | 29.65 (-0.11) | 115 (1.3X) | 27.76 (1.1X) |

Top1 and Top5 shows the accuracy for image classification with (INT8, INT8 - FP32). The models with †, ‡, and § use different accuracy metrics that are shown in Section 4.3. Throughput and latency shows INT8 inference performance and speedup over FP32. Throughput is measured by images per second on large batch size and latency is measured by milliseconds on single batch size.

by 1.6X and 1.5X in average and 2.0X and 2.1X as maximum, respectively. Please note that the convolution improvement on INT8 over FP32 is 1.3X based on HW instructions support Rodriguez et al. (2018) and therefore the latency improvement might be smaller for those non-computation-intensive topologies (e.g., MobileNetV2).

## 4.4 DISCUSSION

To align the model with best accuracy, the above performance in Table 2 does not include INT8 Winograd convolution. We expect to deliver similar performance improvement of Winograd on INT8 as FP32 Lavin & Gray (2016) during our development. Different from previous work Vanhoucke et al. (2011); Sze et al. (2017), we also experiment the first convolution using INT8 than FP32, which shows reasonable accuracy within 1% loss.

Our experimental results also demonstrate the impact of calibration process on accuracy with different sampling iteration, different calibration algorithm, or different scale factor mode. We summarize our findings: **(1)** Channel-wise scaling factors can always deliver better accuracy than single scale factor, especially for depthwise convolution; **(2)** *Direct* algorithm is more effective in most cases than *KL*, while *KL* algorithm can deliver better accuracy than FP32 baseline in some cases; and **(3)** More sampling iterations show more stable dynamic data rage and therefore better accuracy. How to select the optimal calibration strategy is an interesting topic as one of our future directions.

## 5 CONCLUSION

In this paper, we propose the general recipe of INT8 inference and experiment the techniques on a widely-used deep learning framework. We develop an automatic calibration tool for optimized INT8 model generation and demonstrate the effectiveness on 18 CNN models across a wide range of use cases. The inference throughput and latency are improved by 1.6X and 1.5X respectively, while the accuracy loss is minimal within 0.6% to no loss from FP32 baseline. We believe our methodology is general for CNN models and can provide the guide and reference on other frameworks.

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
