# OpenReview forum: "HIGHLY EFFICIENT 8-BIT LOW PRECISION INFERENCE OF CONVOLUTIONAL NEURAL NETWORKS"
_ICLR.cc/2019/Conference_

### Official Review · AnonReviewer1 · 2018-11-03
**provides a white paper of engineering issues for INT8 inference on CPUs that appears to be companion to existing ones for INT8 for GPU**

**Rating:** 4
**Confidence:** 4

**Review:**

This paper reads more like a technical/hardware white paper than a research paper. No real theory is offered, and the results are not really experiments testing hypotheses, but simply reporting the results of their design choices on a various of models.  Thus, it is not clear that this paper is especially suitable for ICLR research track per se.  Furthermore, the calibration method (to find suitable INT8 weights from fp32 ones without further training) appears to be essentially identical to techniques already reported by Nvidia as used by their TensorRT.  The discussion of INT8 for Winograd is something that could have been new and interesting (i.e. this reader has not seen Nvidia discuss this issue previously), but in the end this paper did not offer anything surprising or especially insightful in the brief Section 3.2.2 explaining their approach.  Furthermore, the experiments such as Table 2 do not include Winograd results anyway because that does not give competitive results using INT8, as the authors admit.

---

### Official Review · AnonReviewer2 · 2018-11-03
**More experimental results should be provided**

**Rating:** 4
**Confidence:** 4

**Review:**

This paper try to speedup CNN inference with 8-bit quantization. It is practically useful and may be a nice reference for other developers. But the ideas in this paper are trivial and the motivation is not so convincing.

1) For depthwise convolution in Figure 1(b), the dynamic range differences can be eliminated by batch normalization folding. I doubt the necessity of channel-wise scaling. Besides, the author didn't provide any comparison results of single scaling factor and channel-wise scaling factors for depthwise convolution and traditional convolution layers.
2) For Winograd convolution,  the authors proposed to use scale factors after transformation. They should explain more about this method.
3)Pooling and concatenation support is quite easy. Batch normalization folding is a common practice.
4) The author should explain in more detail on fusing convolution and element-wise operations. Sorry I can't get their point.
5) Calibration results should be provided. The author should also tell readers what hardware is used to evaluate the throughput and latency.
6) Detail results of winograd should be given.

Overall, this paper is in-complete and the authors should add more experimental results and improve their description.

---

### Official Review · AnonReviewer3 · 2018-11-06
**a solid paper, but no much novelty**

**Rating:** 6
**Confidence:** 4

**Review:**

This paper designs a system to automatically quantize the CNN pretrained models. This system contains three main components: 1) different scale factors for channel-wise network; 2) Winograd 8bit quantization; 3) topology wise 8bit operation support. All these three techniques are standard ways to perform model quantization. The work is solid in the sense that 1) as far as I know there is no work actually using all of these quantization schemes, and designs a system to automatically do quantization with additional algorithm support(retrain strategy). 2)  Significantly amount of experiment on quantizing 18 existing widely used CNN models for different applications, e.g., image classification, image segmentation, etc. 3) it reports the actually inference speed up comparing INT 32 to INT 8, although most of the speed up is less than 2.0.

Several questions:

1) What is the difference between retrain and calibration? As mentioned in the paper, the system does not require retrain, but it seems to me that calibration step, e.g., sampling to find the maximum values, etc, is a form of retrain. I think if that is the case, maybe some quantized models with retrain is worthy comparing with.

2) Many quantization techniques are used in the system, are there any conclusions for what techniques are most important for a particular CNN network/application?

3) Are there any new/novel quantization algorithms in the system?

4) The inference speed up is mostly less than 2 times, with some are achieving 2.1 speed up while some are without any speedup. Any reason for that? Also what is the overhead of the inference time?

---

### Meta-Review · Area_Chair1 · 2018-12-16
**Area chair recommendation**

**Confidence:** 5
**Recommendation:** Reject

**Metareview:**

The paper proposes to combine three methods of quantization and apply them to neural network compression.  The methods are known in the literature.  There is a lack of theoretical contribution, and experimental results show variable speedups that may not be competitive with the current state-of-the-art in neural network compression.

The majority of reviewers recommend that this paper be rejected.  The authors have not provided a response.